# Understanding the extent of and limitations to conscientious objection to abortion by health care practitioners: A hermeneutic study

**Valerie Fleming**[1]*, **Lucy Frith**[2], **Clare Maxwell**[1]

**1** Faculty of Health School of Public and Allied Health, Liverpool John Moores University, Liverpool, United Kingdom, **2** Centre for Social Ethics and Policy, School of Law, University of Manchester, Manchester, United Kingdom

* v.fleming@ljmu.ac.uk

## Abstract

The United Kingdom's Abortion Act 1967 has attracted substantial controversy, which has centred not only on the regulation of abortion itself, but also on the extent to which conscientious objection should be permitted. The aim of this study was to examine a range of health-care professionals' views on conscientious objection and identify the appropriate parameters of conscientious objection to abortion. Gadamer's hermeneutic was utilised to frame this study. We conducted semi-structured interviews in two UK locations with 18 pharmacists, 17 midwives, 12 nurses and nine doctors, encompassing a mix of conscientious objectors and non-objectors to abortion. A multi-faceted in-depth data analysis led to the development of a hermeneutic of "respecting self and others". Four major themes of "doing the job", "entrusting to others", "acknowledging institutional power" and "being selective" and 18 subthemes contributed to this overarching theme. The complexity of the responses indicates that there is little consistency within and between each profession. They show that participants who were conscientious objectors were accepted by their colleagues and accommodated without detriment to the service, and that in larger hospitals, such as those where our work was carried out, it is possible to be employed in the service areas that include abortion while still being a conscientious objector. Finally, our results indicate that, by respecting of self and others, each profession should be able to accommodate conscience-based objections where individual practitioners seek to exercise them. Conscientious objectors as well as non-objectors have something to contribute to the ongoing development of the maternity and gynaecological services as abortion is only a small part of the work of these services.

**Data Availability Statement:** All relevant files are available from the LJMU Data Repository (Link: https://doi.org/10.24377/LJMU.d.00000101)

**Funding:** VF, LF and MN received funding of £250,000 to carry out this study from the Economic and Social Research Council (UK). https://www.ukri.org/councils/esrc/ The funder played no role in the study design, data collection and analysis, decision to publish, or preparation of the manuscript

**Competing interests:** The authors have declared that no competing interests exist.

# Background

The Abortion Act 1967, the first legislation on abortion to cover Scotland, England and Wales, has had a significant impact on social norms and healthcare practice. The Act permits abortion on certain grounds concerning foetal or maternal health as certified by two medical practitioners [1]. Abortion is now widely available through the National Health Service (NHS). There was substantial controversy around the introduction of the Act, and it had the power to be socially-divisive within families and communities [2]. One of the main questions that now generates social, professional, and academic controversy is the scope of the right to conscientiously object to participating in abortion under section 4 of the Act [3–7]. There is little scholarly consensus on whether and to what extent conscientious objection should be permitted. Under the Abortion Act, anyone (not only health professionals) is permitted to exercise conscientious objection, while some of the more recently enacted laws restrict the personnel, for example in the Republic of Ireland, which limits conscientious objection to medical practitioners, nurses and midwives [8]. Other countries e.g. Croatia have laws that are specific to each profession [9].

The issue of conscientious objection derives from the ethical controversy over abortion itself. It was recognised when the Act was passed in 1967 that there was not a societal consensus on the ethical acceptability of abortion, and some people would still consider abortion to be immoral and therefore provision was made to allow those who did not agree with the practice to 'opt out', conscientiously object. And the more recent Council of Europe's Resolution 1763 reiterating the right to conscientious objection to abortion in Resolution 1763 "No person, hospital or institution shall be coerced, held liable or discriminated against in any manner because of a refusal to perform, accommodate, assist or submit to an abortion, the performance of a human miscarriage, or euthanasia or any act which could cause the death of a human foetus or embryo, for any reason" [10]. Conscientious objection gives the ethical position of not wanting to participate in abortion practices a legal basis. Fleming et al 's (2018) analysis of reasons for conscientious objection by midwives and nurses noted that the highest number of reasons was grounded in moral, rather than religious reasons [11].

Wicclair has debated the key ethical issues of conscience and integrity in medicine for almost three decades, concluding that there is a need for a more nuanced understanding and analysis of the relevant morals and values and conscientious objection should only be supported if it is underpinned by the core values of medicine. He reiterates this stance in his critique of a proposed rule to protect health professionals' conscience objection in the USA. He argues that legislators should have incorporated a clause to ban invidious discrimination of patients by healthcare professionals that can occur in the name of conscientious objection [12]. Wicclair's position has been supported by writers who suggest that the various rules permitting conscientious objection on the grounds of human rights may have compromised women's right to abortions [13]. Taking note of such problems, it has been argued that in some European countries' laws need to be amended to ensure conscientious objection is not being utilised to undermine women's access to abortion [14]. However, a case study report on four European countries concludes that, although complex, it is possible to accommodate individuals who object to providing abortion-related care, while still ensuring that women have access to legal health care services [3].

At the time of the 1967 Act, abortion was primarily a short surgical procedure in which the primary participant, other than the woman herself, was the surgeon. Nowadays, however, the process is mainly medical, involving many other health professionals such as pharmacists, nurses and, in mid to late term abortions, midwives. Nevertheless, the literature remains focused on doctors. One major study for example surveyed 2000 practising doctors in the

United States out of which 1144 (63%) responded [15]. Its results showed that most doctors believe that it is ethically permissible for doctors to explain their moral objections to patients (63%). A qualitative study of 18 US based doctors, however, suggested that application of conscience objection could only be context dependent thus an overall policy could never be possible [16].

Although some published literature is now reporting studies concerning non-medical health professionals' views or experiences of conscientious objection [17, 18], in Europe there is a death of such studies despite recent legal challenges to laws in Scotland, Croatia and Sweden by midwives who objected to abortion provision on grounds of conscience [19]. A systematic review assessing nurses' and midwives' reasons for conscientious objection showed only nine articles that had discussed the topic, none of which provided empirical evidence [11]. Since the publication of this review, a Canadian study of eight nurses in Canada showed that ethical meaning, whereby their practice was underpinned by moral integrity, ethical knowledge and, when required, courage were important factors when considering conscience-based objections [20]. The main message from a qualitative study carried out in the UK of 20 pharmacists was that, despite differing personal views, pharmacists believed that they should have the right to be able to conscientiously object to abortion [21]. One further US study's researchers interviewed 19 nurses and 31 doctors who worked in labour wards and encountered abortion in the course of their practice about their views on conscientious objection [22]. They concluded that conscience objection is an emerging, iterative process influenced not only by background beliefs of their participants but also by personal and work experiences and the workplace contexts. The aim of the present study, therefore, was to study a range of healthcare professionals' understandings of conscientious objection and its limitations.

## Methods

As the study is concerned with understanding, the hermeneutic philosophy of Hans-Georg Gadamer was utilised throughout this project [23].

A five stage method was applied noting that, in keeping with Gadamer's hermeneutic, these are not consecutive but cyclical, and the final stage of rigour permeates throughout [24]. Stage one of this study resulted in the formulation of two initial questions:

1. What do health professionals understand as constituting 'participation in abortion'?

2. On what grounds should health professionals be permitted to withdraw from provision of abortion services?

### Pre-understandings of the researchers and the reflexive process adopted

Stage two of the research was rooted in Gadamer's notion that every kind of interpretation has roots in a "fore structure of understanding" [25]. Our perceptions were informed by available literature, as well as reports from the popular press and social media. It was thus vital that before and during the study we undertook an examination of our own pre-understandings surrounding conscientious objection to abortion [26].

It was recognised by the research team that the study questions we sought to answer were emotionally and ethically charged and had the potential to be further complicated by the differing views and experiences each individual researcher brought to the project. Prior to our data collection, the research team members audio-recorded a discussion of our own pre-understandings surrounding the subject of conscientious objection to abortion. After several months of data collection we undertook a second audio-recorded discussion, particularly to

examine whether any biases, conscious or otherwise, had influenced our questions or initial analysis. Finally, six months after commencing data analysis, we came together and explored how our pre and altered understandings contributed to and influenced our project, recognising that as Gadamer notes, it is only by consciously assimilating pre-understandings that we can avoid *"the tyranny of hidden prejudices that make us deaf to the language that speaks to us in tradition"* [27]. While we noted some changes to the pre-understandings of one of the team, these did not affect the rigorous analysis process we adopted.

## Recruitment and participants

Ethical approval was granted by the university (anonymised) and ratified by the Health Research Authority (IRAS 246528). Our study was carried out in two large urban areas with specialised women's reproductive health services. The study participants comprised nurses, doctors, midwives and pharmacists. Inclusion criteria required them to be UK registered and currently in one of the two study centres. Participants could be at any career stage, including being newly qualified. The broad nature of the inclusion criteria was a deliberate attempt to capture data, that represented a diversity of views, experiences and perspectives held by the participants.

Participants were recruited via gatekeepers, snowballing and posting information on a local health professional Facebook page. Recruitment was undertaken over one year from February 2019 to February 2020. Health professionals expressing an interest in being interviewed were contacted by the research team to set a date, time and venue of their choice for the interview. Written consent was gained and hard copies stored securely at the university. Participants' data could be withdrawn until integration of the data had commenced. In total we interviewed 18 pharmacists (six objectors, 12 non-objectors), 17 midwives (six objectors, 11 non-objectors), 12 nurses (seven objectors, five non-objectors) and nine doctors (two objectors, six non-objectors and one partial objector).

## Data collection

Dialogue with participants is the third stage in the Gadamerian hermeneutic method. We developed a semi-structured interview schedule comprising open-ended questions and prompts. Interviews lasted from 35 minutes to 1.5 hours.

All interviews were undertaken face to face either in participants' homes, their workplaces, or a neutral venue. Face to face semi-structured interviews were employed to collect data owing to their flexibility [28], ability to explore context [29] and their conversational and dialogical nature, which can be seen as a natural extension of the research participants' world [30]. Four members of the research team undertook the interviews and met regularly to review and reflect upon the data collected. Interviews were transcribed verbatim at which stage potential identifiers were removed.

## Data analysis

Our data analysis was multi-faceted and aligned with Gadamer's hermeneutic, moving from the parts to the whole, the whole expanding as the number of interviews increased. The main strategies we adopted was a thematic analysis of the interviews and their integration into a single "whole", developing mind maps to link various concepts between themes, the derivation of continua to reflect the range of answers to each question with key quotes as evidence and finally deriving a hermeneutic model that painted the picture of the "whole" at the point we stopped analysis. With each strategy we were constantly increasing the dimensions of the

hermeneutic "whole" and building up the greater picture, while working within individual parts of the model so deepening our rigorous and reflexive approach.

The findings presented below reflect each of these strategies.

## Results and discussion

The thematic analysis process led to the development of 23 codes relating to six initial themes: participation, dilemmas, effects of conscientious objection, power, rights and stigma/taboo.

### Development of mind maps

As part of the hermeneutic process of going from the parts to the whole and back, we decided to use mind maps as a tool towards reaching the main hermeneutic themes. Four of them formed the bases of our subsequent discussions. These showed the complexity of the data but ultimately enabled us to see the relationships, not only between our major themes, but also amongst individual concepts. It also facilitated the development of continua, by using the quotes supporting each concept to answer the two main research questions in a two-dimensional way as described below.

### Development of continua

Concurrently with the generation and discussion of the mind maps we embarked upon a process of answering each of the research questions by reflecting on what had been emphasised most strongly by individual participants, rather than considering the whole picture. Out of this we formed a continuum of actions responding to each question and agreed by the research team as well as the advisory group for the study. These continua address the range of opinions presented in relation to each of the questions (see Tables 1 and 2).

### Development of hermeneutic themes

From the above continua, we developed a final question: what are the main influences on balancing rights of women having abortions with rights of health professionals to invoke conscientious objection. By moving backwards and forwards between the different ways of analysing the data, keeping Gadamer's philosophy to the fore and returning to the verbatim words of the participants, we were ultimately able to generate hermeneutic themes, which brought together the key elements of all the above ways of developing a hermeneutic understanding of the data. This is summarised in Fig 1. The main themes are centrally placed with the sub themes surrounding them.

The data obtained were subjected to a rigorous process of analysis as articulated above and included a high level of reflexivity throughout the research process. It resulted in a core theme and four major themes. Each theme is supported by a number of subthemes, which in turn have a firm foundation in the data. Because some subthemes can relate to more than one theme, we included them on each occasion. In the next section we present each of the four

**Table 1. What do health care practitioners understand as constituting 'participation in abortion'?**

| No direct involvement | | | Possible involvement | | | | | Direct involvement | | |
|---|---|---|---|---|---|---|---|---|---|---|
| Consultation/advice | Booking women in | Post abortion care | Referring | Supporting the woman through the process | Dr agrees and signs form | Supervising others | Seeking second signature on form | Inducing the abortion | Performing foeticide | Conducting labour and delivery, performing operation (eg curettage) |

Whole process

**Table 2. On what grounds should health professionals be permitted to withdraw from provision of abortion services?**

| Tolerance | Individual judgement | | | | | Employment issues | | | Intolerance | Law |
|---|---|---|---|---|---|---|---|---|---|---|
| Anything | Religious, moral beliefs | Gestation | Number of previous abortions | Gender selection | Rape / abuse /serious fetal abnormality | Availability of alternative provider | Staffing issues, employability | Patients' rights vs health professionals' | Nothing, part of job | Danger to life |

surrounding themes leading up to the core theme of respecting self and others, a theme that embraced all of the others. In line with Gadamerian hermeneutics, we have presented each of the main themes in the form of gerunds. We have presented the participants' narratives before discussing each of them in relation to other relevant research.

**Findings and integration with literature.** *Respecting self and others 1*: *Doing the job*. This theme encompassed some of poignant emotions that health professionals felt when they managed and balanced their own opinions with the necessity of providing a legal, safe and accessible service to women. Both objecting and non-objecting participants commented that working with women having abortions was not a physically pleasant duty but in today's society it was necessary.

A nurse described her view:

*I don't like doing it [abortion]. I don't know anybody who does like doing it. I don't know anybody who likes being involved in it. But, we do it because it's part of our job and because it's lawful and because we take care of people, regardless of what they come in for* (nurse, non-objector).

Her view was supported by another participant who commented:

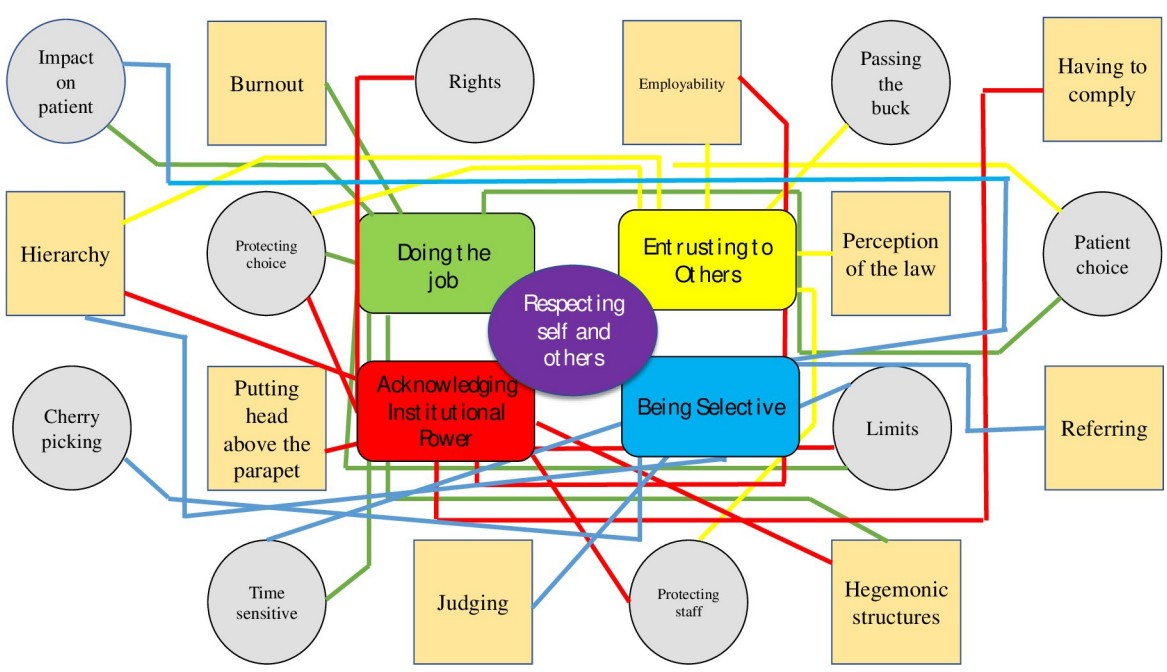

**Fig 1. Final hermeneutic themes.**

*. . .. they will ask you, "Are you okay to go in Room X?" or "Are you okay to go in Room Y?" But I think it's usually because they see whether you can cope with it on that day. Because it is draining, isn't it? It's a different type of work. It's mentally draining* (midwife, objector)

Another participant took a different approach, linking service provision to the needs of the women:

*it's a very selfless and difficult decision that these women are making, and it's incredibly important that we offer that to them* (doctor, non-objector).

Another participant agreed and expanded on this view:

*What I'm trying to do is improve the patient experience. We're trying to do things like with supply of medication, so that when patients do come from a medication point of view their experience is good through what is a very, very emotional thing that they're going to go through. We need to try and make sure there are no bumps in the road for them.* (pharmacist, non-objector).

A doctor summed it up as follows:

*in some ways I struggle with overall because I can see the reasons a woman would choose to have an abortion and why I should not have any say in that but by the same token in respecting that woman's autonomy, I think she has to in some ways respect my autonomy in that I've given things a lot of thought and unfortunately, this is the best way at present that I can reconcile my moral stances and moral difficulties.* (doctor, objector).

Each of these quotes illustrates the complexities of the job of providing abortion related services. It also shows that while some health professionals did not like doing so, they put their personal feelings to one side and simply did the job. This is the first time that empirical evidence has demonstrated this key characteristic of health professionals' ability concurrently to support their patients, their colleagues and themselves in the situation of abortion and backs up what has been reported in the end of life care context [31]. One study outlines the role of health professionals as primarily one of arbitrator between the law and women's requests suggesting that they have to smooth the path for the woman [32]. However, in the present study, although we did not set out to distinguish between different professionals, we found that doctors see themselves in the role of arbitrators while nurses, by contrast, despite the law and whether or not they liked it, felt that they had no choice but to do what they were told [33]. Both objectors and non-objectors in our study acknowledged the primacy of the patient and the need to get the job done but also, for the most part, attempted to accommodate personal beliefs held both by themselves or their colleagues. There were others, however, who held such strong convictions that they could not participate in abortion. This is discussed next.

*Respecting self and others 2: Entrusting to others*. The second theme of "entrusting to others" developed directly from health professionals' thoughts on their rights to opt out of providing care but acknowledging that in some cases they were uncertain about the requirement to find a colleague to take over from them, and the extent of this requirement.

One participant stated blandly:

*As long as they're still seen to, you've done your part in putting that patient's needs first but you've not compromised your beliefs* (pharmacist, objector).

Some participants seemed to be developing their ideas and thoughts during the interview, with one midwife acknowledging:

*I think people should have the right themselves to decide what level of involvement they want. Something that I deem acceptable to be involved while someone else might think, "I wouldn't even want to be involved in that level, and I wouldn't want to care for them post-procedure." So I think it is an individual's right as a human being, even though you are a health professional, that you should still have your own choice and decision-making process whether you do or don't want to be involved in something* (midwife, objector).

A nurse however, did not offer solutions but felt that as a profession their own wellbeing had somehow been neglected:

*. . .like nursing is stressful anyway and if you have your conscientious objection overruled then I think that can have a long-term impact on you as a professional and your emotional wellbeing as well, because I think sometimes nurses and that get forgotten in the mix. You know, you've got to care for the carers basically, and that's physically, spiritually and emotionally* (nurse, non-objector).

A doctor suggested:

*I think the way the health system is set up at the minute, that there is enough scope to protect healthcare providers who are not comfortable to be involved in termination of pregnancy and there are enough systems in place that designated teams can look after patients who are willing to provide the care* (doctor, non-objector).

Conversely a midwife stated:

*And you don't have to say, "I'm not doing that," in handover, but you could, even if you get handed it, go and speak to the shift leader quietly, to say, "I don't agree with it and I don't want to do that, can we swap, I will take somebody else's caseload?" I just think that is probably the only way it could be managed.* (midwife, non-objector).

The above data show that whether or not participants had identified themselves as "objectors", they were generally accommodating of colleagues' wishes not to be involved. A sense of respect also permeated participants' expressions of willingness to refer women for whom they could not provide a service, to others. Both the General Medical Council and General Pharmaceutical Council provide explicit guidelines for their members in which they outline the duty of their registrants who have a conscientious objection to refer to other practitioners who will provide the service [34, 35]. The Nursing and Midwifery Council (2020) obliges its registrants to do likewise. In practice [36], however, this is not so easy for nurses and midwives, who work together in teams, sometimes with varying degrees of experience. It therefore becomes more difficult for practitioners to simply swap allocated workloads with colleagues, so they are obliged to speak quietly to their managers. For the doctors in the study, this did not seen to be an issue at all. Those who voice an objection during their specialist training can opt out of completing a (normally compulsory) module on termination of pregnancy. Only one participant, who was an objector, felt that their objection could impose burdens on colleagues. This was not supported by other doctors in the study most of whom were non-objectors and were comfortable providing support for their objecting colleagues. Others suggest that reasonable

accommodation is possible, or argue that conscientious objection should be accommodated [37, 38]. Both of them, however, only argue only on behalf of doctors. In our study, however, some pharmacists who worked in community settings also felt able to make autonomous decisions and were able to object, although referral to a colleague or another practice had to be made by objectors in accordance with their regulatory body [39]. While the law affords the same protection to other health professionals, as mentioned throughout, we have shown this to be somewhat more difficult to actualise, and often is not fully accommodated.

*Respecting self and others 3*: *Acknowledging institutional power*. The third theme deals with some of the issues concerning resourcing in a large organisation such as National Health Service hospitals while coming down to the micro level that deals with individual employees' preferences. Balancing the needs of the organisation with those of the individual is something that was considered by participants from each profession.

A midwife who was seriously considering her future in providing intrapartum care noted:

*I probably wouldn't be senior enough on shift, when I worked as a midwife, to be asked to do that room [with late abortions] but the more senior you got, you knew that was going to become something that you would have to participate in, but that I wouldn't have participated in* (midwife, objector).

A nurse added:

*a person needs to have a lot of strength to be able to stand up and say no to certain things and they are afraid of the outcome afterwards and if they participate in that against their own emotions then they're going to pay the price because it's going to be on their mind and in their hearts and it can cause illness for them as-well* (nurse, non-objector).

Another nurse recounts: *I had to go down to the pharmacy and collect this medication. . ..a phial, which I refused to do and I was the ordered to go. . ..I refused again. . ..they got the phial; the surgeon who did the procedure then threatened me . . ..he physically threatened me* (nurse, objector).

A midwife, however, stated:

*[they are] agreeing so they'll get a job. A lot of people do that. . ..you're not guaranteed a job at the end of the first three years, so it makes it easier if you agree with what they want you to do to get that job.* (midwife, non-objector).

A doctor described the hierarchy in a pre-abortion clinic:

*whilst I'm seeing the next one, the nurses would have to try and walk around and find somebody that would do a second signature. And, again, if we're thin on the ground of people to find, and those people that are in are. . . We had nobody to sign the document* (doctor, non-objector).

A pharmacist was able to hide in the hierarchical structure:

*We don't actually see the prescriptions. It's very rare that a prescription would ever come via pharmacy. We're just ensuring the ward areas have the stock for someone else to have to make that decision* (pharmacist, objector).

The NHS was thus acknowledged by all as a hierarchy in which some people seemed to have more rights than others. Nurses and midwives juggled the act of conscientiously objecting with the consequences of putting their heads above the parapet. Many of them are employed on short-term contracts, and while abortion care remains a small part of the work of gynaecology departments, some felt by refusing to participate in it, they might forfeit their chances of a permanent post or of internal promotion. Such was the case in Sweden, with a nurse, who, on completion of a post graduate midwifery course, was refused a post at three hospitals because of her stance on conscientious objection to abortion and eventually brought a case to the European Court of Human Rights which declined to admit it [19]. Another researcher noted that if nurses do not have the freedom to exercise their consciences by refusing to participate in practices they believe to be seriously wrong, they risk fracturing their fundamental moral well-being [20, 40]. Objectors, however, found ways in which they could quietly voice this but for nurses in particular it was difficult without upsetting the shift pattern of the day.

Participants also observed that a certain degree of strength may be required for objectors to stand up against a system in which they could be stigmatised. Sawicki, however, notes that at times a person's conscience is so powerful that it compels them to act against the dominant trends by feeling bound by volitional necessity [41]. The nurse who reported being physically threatened for objecting and the midwife who felt that their days in intrapartum care were numbered, believed that this was due to their stated objection to providing abortion care. This is in direct contrast to other research, which maintains that negative attitudes prevail towards health professionals who do provide abortion services rather than toward those who refuse to provide it [21].

*Respecting self and others 4: Being selective.* The final theme involved discussion around the elements of the process which health professionals could object and amongst participants there was general agreement, as one midwife put it, about the elements to which objectors could opt out:

One midwife said:

*If you don't want to answer a buzzer in that room, that's fine, don't. I suppose, because we work in such a big unit, that's okay. There are 12 of us. If one person chooses to not answer a buzzer, it's not a big deal, that's fine, someone else will* (midwife, non-objector)

Another participant noted that objecting was sometimes a case of "shifting sands":

*I don't think you object to it all or accept it all. . . .I mean I know people who are at varying degrees of where they sit on things in terms of gestations and I get that. I get that different things happen in your life and you just mightn't be able to do that at that point* (doctor, non-objector).

A pharmacist made a similar point, saying:

*I just think it just depends on the healthcare professional on how they want to deal with it. I mean, they should have the right to say I do not want to talk to you, I do not want to answer any more questions you have* (pharmacist).

One participant was adamant that the only thing to which doctors should be allowed to object was signing the form to permit abortions to take place:

*I've often been asked to sign the medication to induce the abortion. I've said that I would prefer not to sign, I would prefer not to prescribe the medication, and there are often other doctors that are available. But I am sometimes concerned that my refusal may put other people not under pressure, but I do find it difficult every time I'm asked to have to justify myself and there are undertones* (doctor, objector).

A midwife brought a different perspective:

*I personally would object to anything that involved being involved in the foeticide [injecting a lethal substance into the foetus before inducing labour] side of things, so actually ending the life. But the actual care post, once that has happened, and the delivery and postnatal support, is something that I would happily provide to women* (midwife, objector).

The final theme has shown the broad range of participants' views as to what aspects of the abortion process to which they could object. Additionally for some the situation might vary from day to day or case to case. "Being selective" generated negative feelings amongst participants but no consistency as to the way forward. For example, some participants felt uncomfortable in being involved with so called social abortions. Many more, however, were less relaxed about providing services as gestation increased and still more when foeticide was required. Some participants reported that objection should be "all or nothing", although most took a more nuanced approach, with some arguing that they should be at liberty make an individual judgement on each occasion they were required to. Our continua also showed that there were several shades of grey in between the extremes of "all or nothing". Commenting in this area have also suggested that by applying the law, doctors should be afforded the discretionary space to participate or not and to the degree to which they are comfortable [42]. Sepper argued from a different perspective, proposing that conscience "may be experienced retrospectively, generating guilt or regret, or prospectively, generating a sense that failure to resolve these conflicting demands will risk one's sense of self" [43]. By this she means that one's conscience may come to the fore only following the actions rather than preceding them and so people's own positions may change evolve over time.

## Conclusions, limitations and recommendations

Our research has offered the first in-depth analysis of the beliefs about conscientious objection of health professionals from disciplines which have previously has previously received little scholarly attention. It represents is the largest qualitative empirical study, and the first in Europe, on four different professional groups' perceptions of conscientious objection to abortion. Throughout our research, we have been true to the Gadamerian principle that understanding can never be complete and this is reflected by in our use of the gerund as we named each theme. Additionally, by constantly addressing our own pre-understandings, coming from different standpoints, and using them to challenge our findings at all stages during the analysis process we ensured rigour thereby achieving the fifth and final stage of the method we adopted. As such this study offers new insights into practising health professionals' understandings of the multi-faceted and complex nature of conscientious objection to abortion.

While not seeking to differentiate between the four health professional disciplines, it became obvious that nurses in particular felt that they were not in a position to make conscientiously such objections in the way doctors were. Additionally, it seemed that the law was not fully understood by all participants, some of whom intimated that it was not possible to exercise their legal right to conscientiously object to participate in abortions.

The complexity of the responses also indicates that there is little consistency within each profession although generally participants who were objectors were accepted by their colleagues and accommodated without detriment to the service and that in larger hospitals, such as those where our work was carried out, it is possible to be employed in the service areas that include abortion while still being an objector.

Our findings also indicated very divergent views as to what constitutes conscientious objection and in some cases showed uncertainty and confusion such as how to go about notifying employers and colleagues of a conscientious objection. Our research questions therefore did not yield definitive answers and so merit further research with a larger sample, taking our results and developing a questionnaire with which to quantify responses.

Although this was a large qualitative study, the results cannot be generalised but they may be applicable to health professionals working in similar settings, such as end of life care. The proposed questionnaire could also be used to assess the situation in smaller district general hospitals.

Finally, our results indicate that, by respecting of self and others, health professionals should be able to accommodate conscience-based objections where this is sought by individual practitioners. Objectors as well as non-objectors have something to contribute to the ongoing development of maternity and gynaecological services, as abortion is only a small part of the work of these services.

## Author Contributions

**Conceptualization:** Valerie Fleming, Lucy Frith.

**Data curation:** Valerie Fleming, Lucy Frith, Clare Maxwell.

**Formal analysis:** Valerie Fleming, Clare Maxwell.

**Funding acquisition:** Valerie Fleming.

**Investigation:** Valerie Fleming, Clare Maxwell.

**Methodology:** Valerie Fleming.

**Project administration:** Valerie Fleming.

**Writing – original draft:** Valerie Fleming, Lucy Frith, Clare Maxwell.

**Writing – review & editing:** Valerie Fleming, Lucy Frith, Clare Maxwell.

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
