## [Decision Letter · Decision Letter 0]

27 Mar 2023

PONE-D-23-03045Understanding the extent of and limitations to conscientious objection to abortion by health care practitioners: a hermeneutic studyPLOS ONE

Dear Dr. Fleming,

Thank you for submitting your manuscript to PLOS ONE. After careful consideration, we feel that it has merit but does not fully meet PLOS ONE’s publication criteria as it currently stands. Therefore, we invite you to submit a revised version of the manuscript that addresses the points raised during the review process.

Please see the comments from two reviewers below. Please note that there are no word limits for PLOS ONE, but we encourage you to consider the reviewer's suggestions for length and content of different sections. In the interest of reproducibility, if you choose to abbreviate the Methods section, please ensure that the full methodology is included as a Supplementary file. You may also consider presenting the full Methods at the end of the manuscript, as PLOS ONE does not require the Methods to be presented before the Results section.

We look forward to receiving your revised manuscript.

Kind regards,

Hanna Landenmark

Staff Editor

PLOS ONE

Journal Requirements:

https://bmcmedethics.biomedcentral.com/articles/10.1186/s12910-018-0268-3

In your revision ensure you cite all your sources (including your own works), and quote or rephrase any duplicated text outside the methods section. Further consideration is dependent on these concerns being addressed.

Reviewers' comments:

Reviewer's Responses to Questions

**Comments to the Author**

1. Is the manuscript technically sound, and do the data support the conclusions?

Reviewer #1: Partly

Reviewer #2: Partly

2. Has the statistical analysis been performed appropriately and rigorously? 

Reviewer #1: N/A

Reviewer #2: I Don't Know

3. Have the authors made all data underlying the findings in their manuscript fully available?

Reviewer #1: Yes

Reviewer #2: No

4. Is the manuscript presented in an intelligible fashion and written in standard English?

Reviewer #1: Yes

Reviewer #2: Yes

5. Review Comments to the Author

Reviewer #1: This is very interesting empirical research that addresses an important topic. However, the paper reads as a rough first draft rather than a final, polished article.

While methodology is important, I don’t think that the detailed description of the methodology is necessary. It should not take until page 8 to get to the substance of the article. I recommend condensing the content of the article up to page 8 which is essentially an overly detailed description of the methodology.

While the researchers have clearly engaged in a lot of work figuring out what are the themes of the data, these themes are presented as a series of quotes from the interviews and are lacking in analysis. Presenting the quotes is step 1, but the authors then need to go on to discuss and analyse those quotes and how they relate to the identified theme.

I think it would make sense to restructure the article so that the discussion is included in the analysis of the themes, rather than being presented separately. Otherwise the setting out of the themes is just a series of quotes. Alternatively, the authors may wish to present a more traditional version of a results section followed by discussion.

The way that the views of the authors is presented is problematic. On the one hand, the authors frame the research as being about 2 questions:

1. What do health professionals understand as constituting ‘participation in abortion’?

2. On what grounds should health professionals be permitted to withdraw from provision of

abortion services?

But the article does not clearly address these two questions. Further, the authors seem to adopt a position on whether conscientious objection should be allowed and in what circumstances. It is not in and of itself problematic for the authors to adopt a position, but they don’t admit to doing so in the article – their position is presented as fact rather than opinion and is not clearly based on the data.

Reviewer #2: Overall, the manuscript contributes much-needed empirical data on the topic of health practitioners' CO to abortion. More background context is needed to understand the results of the study, and the authors should demonstrate more rigour in places (particularly given the ethically controversial nature of abortion, and CO).

Background:

More context is needed about the current UK legal framework as it relates to CO to abortion. Is CO to abortion protected in legislation? If so, which health practitioners does the clause apply to? What are the legal requirements for claiming CO? And what are the objecting practitioner’s obligations e.g. declaring the CO to the patient, referring on etc?

If CO is legally protected, then this would surely have implications for whether health services are required to accommodate it?

Also, further detail is needed about the context of abortion provision in the UK. For example, are abortions all publicly funded? Or does the private sector play a role? What is the pathway for having an abortion e.g. explanation of the form requiring two signatures?

p. 4:

“Little has been written about other health professionals’ views or experiences of conscientious

objection , despite recent legal challenges to laws in Scotland, Croatia and Sweden by midwives who

objected to abortion provision on grounds of conscience”

There are other empirical studies that focus on or include non-doctor CO to abortion such as:

• Awonoor-Williams et al Exploring conscientious objection to abortion among health providers in Ghana. International Perspectives on Sexual and Reproductive Health 2020 Vol. 46 Pages 51-59

• Davidson et al. Religion and conscientious objection: A survey of pharmacists’ willingness to dispense medications. Social Science&Medicine71(2010)161e165

• Harries et al 2014 Conscientious objection and its impact on abortion service provision in South Africa: a qualitative study. Harries et al. Reproductive Health 2014, 11:16 http://www.reproductive-health-journal.com/content/11/1/16

• Ko et al et al An ethical issue: nurses’ conscientious objection regarding induced abortion in South Korea.

• Lee et al. Mifepristone (RU486) in Australian pharmacies: the ethical and practical challenges. Contraception 91 (2015) 25–30.

Methods:

p. 5: “Finally, six months after commencing data analysis, we came together and explored how our pre and altered understandings contributed to and influenced our project, recognising that as Gadamer notes, it is only by consciously assimilating preunderstandings that we can avoid “the tyranny of hidden prejudices that make us deaf to the language that speaks to us in tradition” (Gadamer, 2006, p. 239).”

It’s good you have formally reflected on your understandings of CO and how they have contributed and influenced the project. Please provide some description about these understandings and how they influenced your data collection and analysis.

Recruitment and participants: Good sample size with both objectors and non-objectors. Range of health practitioners included.

Data collection: Please provide the interview guide as an appendix or supplemental material.

Tables 1 and 2 are not integrated with the main body of the text. How did you ensure rigour in developing the continua?

P. 13 – information about referral requirements should be part of the Background

Referencing software needs attention – incorrect bracketing of in-text citations throughout

Conclusions:

“Our research represents the largest qualitative empirical study and the first in Europe on the subject of health professionals’ conscientious objection to abortion.”

Agree it is a large qualitative, empirical study. But I’m not sure that it’s the first in Europe? What about these?

• De Zordo (2018) From women’s ‘irresponsibility’ to foetal ‘patienthood’:

Obstetricians-gynaecologists’ perspectives on abortion and its stigmatisation in Italy and Cataluña. GLOBAL PUBLIC HEALTH, 2018 VOL. 13, NO. 6, 711–723

• Nordberg (2014) Conscientious objection to referrals for abortion:pragmatic solution or threat to women’s rights? Medical Ethics 2014, 15:15 http://www.biomedcentral.com/1472-6939/15/15

• Toro-Flores (2019) Opinions of nurses regarding conscientious objection. Nursing Ethics, 2019, Vol. 26(4) 1027–1038

Conclusion

Respecting self and others shows that participants who were objectors were accepted by

their colleagues and accommodated without detriment to the service, and that in larger hospitals…

The result below sounds like a detriment to the service?

A doctor described the hierarchy in a pre-abortion clinic:

whilst I’m seeing the next one, the nurses would have to try and walk around and find somebody that

would do a second signature. And, again, if we’re thin on the ground of people to find, and those

people that are in are… We had nobody to sign the document (doctor, non-objector). P. 11

“Our study was carried out in two large urban areas with specialised women’s reproductive health

services.” This context would be useful in the methods section. Also, perhaps this could be a reason that difficulties with abortion access stemming from CO did not feature in this study?

Conclusion could be more concise.

6. PLOS authors have the option to publish the peer review history of their article (what does this mean?). If published, this will include your full peer review and any attached files.

Reviewer #1: No

Reviewer #2: No

---

## [Author Response · Author response to Decision Letter 0]

21 Apr 2023

Dear editor and reviewers

Thank you so much for your sharp eyes and critical comments. On behalf of the team, I enclose a response to your comments as well as the revised manuscript. For ease of reading, this is in a word doc...file 7. 

Kind regards

Valerie

1. Please ensure that your manuscript meets PLOS ONE's style requirements, including those for file naming. Thank you, done 

2. We noticed you have some minor occurrence of overlapping text with the following previous publication(s), which needs to be addressed. In your revision ensure you cite all your sources (including your own works), and quote or rephrase any duplicated text outside the methods section. Further consideration is dependent on these concerns being addressed. We’ve been through this and cannot find anything that has not been cited including team members’ previous publications. Please advise is we’ve overlooked something. 

 Sorry, sorted. The mistake came about because additional funding was received during the pandemic.

3. Have the authors made all data underlying the findings in their manuscript fully available?

Reviewer #1: Yes 

Reviewer #2: No All data is available on the link listed. We fail to see why “partly” has been written here. 

Reviewer #1: This is very interesting empirical research that addresses an important topic. However, the paper reads as a rough first draft rather than a final, polished article.

 Sorry, that is how you feel. A lot of consultation and work went into the presentation. It was definitely NOT a rough first draft. However, there is always room for improvement and we have gone through it again making subtle and not so subtle changes. 

While methodology is important, I don’t think that the detailed description of the methodology is necessary. It should not take until page 8 to get to the substance of the article. I recommend condensing the content of the article up to page 8 which is essentially an overly detailed description of the methodology. This was difficult because the other reviewer wanted more detail. In fact the results began on page 5 (commencing on page 2) not page 8 of the Pdf submitted. We have, however, trimmed the methodology back a bit. 

While the researchers have clearly engaged in a lot of work figuring out what are the themes of the data, these themes are presented as a series of quotes from the interviews and are lacking in analysis. Presenting the quotes is step 1, but the authors then need to go on to discuss and analyse those quotes and how they relate to the identified theme.

I think it would make sense to restructure the article so that the discussion is included in the analysis of the themes, rather than being presented separately. Otherwise the setting out of the themes is just a series of quotes. Alternatively, the authors may wish to present a more traditional version of a results section followed by discussion.

 Yes, we agree. We have integrated the findings and discussion as it seems to read better that way. 

The way that the views of the authors is presented is problematic. On the one hand, the authors frame the research as being about 2 questions:

1. What do health professionals understand as constituting ‘participation in abortion’?

2. On what grounds should health professionals be permitted to withdraw from provision of abortion services? But the article does not clearly address these two questions. 

 Again, food for thought. You are right. We have amended our conclusions to reflect this. 

Further, the authors seem to adopt a position on whether conscientious objection should be allowed and in what circumstances. It is not in and of itself problematic for the authors to adopt a position, but they don’t admit to doing so in the article – their position is presented as fact rather than opinion and is not clearly based on the data. We can’t see where we have presented our own positions. In fact we all held different ones. We have updated the detailed section on our own pre-understandings. 

Reviewer #2: Overall, the manuscript contributes much-needed empirical data on the topic of health practitioners' CO to abortion. More background context is needed to understand the results of the study, and the authors should demonstrate more rigour in places (particularly given the ethically controversial nature of abortion, and CO).

Background:

More context is needed about the current UK legal framework as it relates to CO to abortion. Is CO to abortion protected in legislation? If so, which health practitioners does the clause apply to? What are the legal requirements for claiming CO? And what are the objecting practitioner’s obligations e.g. declaring the CO to the patient, referring on etc?

If CO is legally protected, then this would surely have implications for whether health services are required to accommodate it?

Also, further detail is needed about the context of abortion provision in the UK. For example, are abortions all publicly funded? Or does the private sector play a role? What is the pathway for having an abortion e.g. explanation of the form requiring two signatures?

p. 4:

“Little has been written about other health professionals’ views or experiences of conscientious

objection , despite recent legal challenges to laws in Scotland, Croatia and Sweden by midwives who

objected to abortion provision on grounds of conscience”

There are other empirical studies that focus on or include non-doctor CO to abortion such as:

• Awonoor-Williams et al Exploring conscientious objection to abortion among health providers in Ghana. International Perspectives on Sexual and Reproductive Health 2020 Vol. 46 Pages 51-59

• Davidson et al. Religion and conscientious objection: A survey of pharmacists’ willingness to dispense medications. Social Science&Medicine71(2010)161e165

• Harries et al 2014 Conscientious objection and its impact on abortion service provision in South Africa: a qualitative study. Harries et al. Reproductive Health 2014, 11:16 https://ddec1-0-en-ctp.trendmicro.com:443/wis/clicktime/v1/query?url=http:%2f%2fwww.reproductive%2dhealth%2djournal.com%2fcontent%2f11%2f1%2f16&umid=396fdfda-2ba2-44f8-8ab6-8053d1f379e8&auth=6b639a990a359ff1d6cc8761081d57748ce3c81e-2664dc5e8bc258c533ca3f73418ec6a8cb368adf

• Ko et al et al An ethical issue: nurses’ conscientious objection regarding induced abortion in South Korea.

• Lee et al. Mifepristone (RU486) in Australian pharmacies: the ethical and practical challenges. Contraception 91 (2015) 25–30.

. See comments above to the other reviewer. We have added some and removed some and we hope it is now satisfactory to you both. 

Methods:

p. 5: “Finally, six months after commencing data analysis, we came together and explored how our pre and altered understandings contributed to and influenced our project, recognising that as Gadamer notes, it is only by consciously assimilating preunderstandings that we can avoid “the tyranny of hidden prejudices that make us deaf to the language that speaks to us in tradition” (Gadamer, 2006, p. 239).”

It’s good you have formally reflected on your understandings of CO and how they have contributed and influenced the project. Please provide some description about these understandings and how they influenced your data collection and analysis. See comments to other reviewer. I think we’ve now addressed this. 

Recruitment and participants: Good sample size with both objectors and non-objectors. Range of health practitioners included.

 Thank you

Data collection: Please provide the interview guide as an appendix or supplemental material.

Tables 1 and 2 are not integrated with the main body of the text. 

 Done, thanks.

How did you ensure rigour in developing the continua?

 Sentence added, thank you. 

P. 13 – information about referral requirements should be part of the Background

Referencing software needs attention – incorrect bracketing of in-text citations throughout

 Both these done

Conclusions:

“Our research represents the largest qualitative empirical study and the first in Europe on the subject of health professionals’ conscientious objection to abortion.”

Agree it is a large qualitative, empirical study. But I’m not sure that it’s the first in Europe? What about these?

• De Zordo (2018) From women’s ‘irresponsibility’ to foetal ‘patienthood’:

Obstetricians-gynaecologists’ perspectives on abortion and its stigmatisation in Italy and Cataluña. GLOBAL PUBLIC HEALTH, 2018 VOL. 13, NO. 6, 711–723

• Nordberg (2014) Conscientious objection to referrals for abortion:pragmatic solution or threat to women’s rights? Medical Ethics 2014, 15:15 https://ddec1-0-en-ctp.trendmicro.com:443/wis/clicktime/v1/query?url=http:%2f%2fwww.biomedcentral.com%2f1472%2d6939%2f15%2f15&umid=396fdfda-2ba2-44f8-8ab6-8053d1f379e8&auth=6b639a990a359ff1d6cc8761081d57748ce3c81e-7030e6466c730d252a18813d7ba0d805b0ff6b11

• Toro-Flores (2019) Opinions of nurses regarding conscientious objection. Nursing Ethics, 2019, Vol. 26(4) 1027–1038

 You are right. Sentence has been amended. 

Thanks for the references. Most of them we already had but the de Zordo was new and really interesting. It’s a shame it was only on drs! 

Conclusion

Respecting self and others shows that participants who were objectors were accepted by

their colleagues and accommodated without detriment to the service, and that in larger hospitals…

The result below sounds like a detriment to the service?

A doctor described the hierarchy in a pre-abortion clinic:

whilst I’m seeing the next one, the nurses would have to try and walk around and find somebody that

would do a second signature. And, again, if we’re thin on the ground of people to find, and those

people that are in are… We had nobody to sign the document (doctor, non-objector). P. 11

“ You have a good point. Thank you. Amended. We are going to create another article specifically about nurses in this regard.

Our study was carried out in two large urban areas with specialised women’s reproductive health

services.” This context would be useful in the methods section. Also, perhaps this could be a reason that difficulties with abortion access stemming from CO did not feature in this study?

 Possibly but it’s these hospitals that offer the range of abortion services.

Conclusion could be more concise Thank you. Done.

---

## [Decision Letter · Decision Letter 1]

25 Oct 2023

PONE-D-23-03045R1Understanding the extent of and limitations to conscientious objection to abortion by health care practitioners: a hermeneutic studyPLOS ONE

Dear Dr. Fleming,

Thank you for submitting your manuscript to PLOS ONE. After careful consideration, we feel that it has merit but does not fully meet PLOS ONE’s publication criteria as it currently stands. Therefore, we invite you to submit a revised version of the manuscript that addresses the points raised during the review process. The article is implemented, however further important insights are needed and clarity implementation is also needed through copyediting, typos fix and grammar check.

We look forward to receiving your revised manuscript.

Kind regards,

Andrea Cioffi

Academic Editor

PLOS ONE

Reviewers' comments:

Reviewer's Responses to Questions

**Comments to the Author**

1. If the authors have adequately addressed your comments raised in a previous round of review and you feel that this manuscript is now acceptable for publication, you may indicate that here to bypass the “Comments to the Author” section, enter your conflict of interest statement in the “Confidential to Editor” section, and submit your "Accept" recommendation.

Reviewer #2: (No Response)

Reviewer #3: (No Response)

2. Is the manuscript technically sound, and do the data support the conclusions?

Reviewer #2: Yes

Reviewer #3: Partly

3. Has the statistical analysis been performed appropriately and rigorously? 

Reviewer #2: Yes

Reviewer #3: I Don't Know

4. Have the authors made all data underlying the findings in their manuscript fully available?

Reviewer #2: Yes

Reviewer #3: Yes

5. Is the manuscript presented in an intelligible fashion and written in standard English?

Reviewer #2: Yes

Reviewer #3: Yes

6. Review Comments to the Author

Reviewer #2: Revisions have improved the manuscript. Given members of the author team held different personal positions on CO, I think this should be stated explicitly in the paper. It is a strength of the author team and of the paper.

Reviewer #3: The authors deal with a particularly hot topic and very much addressed in literature from various points of view. Therefore, it is inevitable that my opinions and comments consider the originality of the content.

Firstly, in my opinion, there is one key point missing from the introduction, namely an argument on bioethical issues of conscientious objection. On which rights does conscientious objection rest? what does it aim to protect? Why is this right granted even in circumstances where it concerns the psychophysical health of patients?

I find that such a premise is fundamental to give substance to the introduction which, at present, is too superficial and is far from taking stock of the real issue. In addressing issues of this nature, we should start from a solid bioethical argument, indispensable to be able to effectively argue the answers provided by health professionals interviewed.

In fact, even on such occasions, the authors' arguments are rather hasty. Yet the results of the questionnaires are quite interesting so it would be a shame if they were lost.

I invite the authors to review these aspects trying to unbalance themselves a little more in the discussions and giving a more critical imprint to the article, also considering the great experience of the authors on the subject.

I hope that authors can find my comment motivating, as I do myself with other reviewers when I am in their position.

Many thanks to the Editor for the opportunity to review this article for such a prestigious journal.

7. PLOS authors have the option to publish the peer review history of their article (what does this mean?). If published, this will include your full peer review and any attached files.

Reviewer #2: No

Reviewer #3: No

---

## [Author Response · Author response to Decision Letter 1]

21 Nov 2023

Reviewer #2: Revisions have improved the manuscript. Given members of the author team held different personal positions on CO

I think this should be stated explicitly in the paper. It is a strength of the author team and of the paper. 

Thank you, we have strengthened this. It’s difficult at times to differentiate what is necessary to say and what should not be said but I certainly do not want to be accused by readers or being a “CO club”!

Reviewer #3: The authors deal with a particularly hot topic and very much addressed in literature from various points of view. Therefore it is inevitable that my opinions and comments consider the originality of the content.

Firstly in my opinion there is one key point missing from the introduction namely an argument on bioethical issues of conscientious objection. On which rights does conscientious objection rest? what does it aim to protect? Why is this right granted even in circumstances where it concerns the psychophysical health of patients?

I find that such a premise is fundamental to give substance to the introduction which at present is too superficial and is far from taking stock of the real issue. In addressing issues of this nature we should start from a solid bioethical argument indispensable to be able to effectively argue the answers provided by health professionals interviewed.

In fact even on such occasions the authors' arguments are rather hasty. 

It is indeed a hot topic and thank you for the critique. We all appreciate it. We have each had a go at this and our ethicist on the authors has particularly strengthened it. The simple answer perhaps is in your own question re the health of patients. I think the objectors felt that they had two patients….one who couldn’t speak for themselves. However, that’s just FYI as we could go down a whole different pathway. It’s really great that we have an ethicist on the team who is able to move into empirical research and we hope this article will have some “punch” with clinical practitioners.

Yet the results of the questionnaires are quite interesting so it would be a shame if they were lost. 

Just a bit of puzzlement here…we didn’t use questionnaires. 

I invite the authors to review these aspects trying to unbalance themselves a little more in the discussions and giving a more critical imprint to the article also considering the great experience of the authors on the subject. 

Thank you, we’ve tried our best!

I hope that authors can find my comment motivating as I do myself with other reviewers when I am in their position.

Indeed we do and it would be good to be able to have more debate in the future. Thank you for coming on board as a last minute substitution in the reviewers.

---

## [Decision Letter · Decision Letter 2]

2 Jan 2024

Understanding the extent of and limitations to conscientious objection to abortion by health care practitioners: a hermeneutic study

PONE-D-23-03045R2

Dear Dr. Fleming,

We’re pleased to inform you that your manuscript has been judged scientifically suitable for publication and will be formally accepted for publication once it meets all outstanding technical requirements.

Kind regards,

Andrea Cioffi

Academic Editor

PLOS ONE

Additional Editor Comments (optional):

No further revisions are necessary.

Reviewers' comments:

Reviewer's Responses to Questions

**Comments to the Author**

1. If the authors have adequately addressed your comments raised in a previous round of review and you feel that this manuscript is now acceptable for publication, you may indicate that here to bypass the “Comments to the Author” section, enter your conflict of interest statement in the “Confidential to Editor” section, and submit your "Accept" recommendation.

Reviewer #3: (No Response)

2. Is the manuscript technically sound, and do the data support the conclusions?

Reviewer #3: (No Response)

3. Has the statistical analysis been performed appropriately and rigorously? 

Reviewer #3: (No Response)

4. Have the authors made all data underlying the findings in their manuscript fully available?

Reviewer #3: (No Response)

5. Is the manuscript presented in an intelligible fashion and written in standard English?

Reviewer #3: (No Response)

6. Review Comments to the Author

Reviewer #3: I apologize for the accidental oversight with which I called the interviews "questionnaires".

The questions I asked the authors did not require a personal answer, but were intended as examples, ideas to introduce critical reflections into the manuscript.

I still think that the interesting results could have been better argued by the authors.

I would like to stress that adding something new to scientific literature, especially in this area, does not just mean proposing a type of study with an innovative design. It can also mean doing a study quite similar to those already in the literature, but with a brilliant and new critical approach.

Having said that, I believe that the revisions have sufficiently implemented the article, which, in my opinion, is publishable because of its originality.

I thank the authors for the debate and the Editor for the opportunity and the confidence shown in me.

7. PLOS authors have the option to publish the peer review history of their article (what does this mean?). If published, this will include your full peer review and any attached files.

Reviewer #3: No

---

## [Editor Report · Acceptance letter]

14 Feb 2024

PONE-D-23-03045R2 

PLOS ONE

Dear Dr. Fleming, 

I'm pleased to inform you that your manuscript has been deemed suitable for publication in PLOS ONE. Congratulations! Your manuscript is now being handed over to our production team.

Kind regards, 

on behalf of

Dr. Andrea Cioffi 

Academic Editor

PLOS ONE